# Intermediate Hair Follicles from Patients with Female Pattern Hair Loss Are Associated with Nutrient Insufficiency and a Quiescent Metabolic Phenotype

**DOI:** 10.3390/nu14163357

**Published:** 2022-08-16

**Authors:** Ilaria Piccini, Marta Sousa, Sabrina Altendorf, Francisco Jimenez, Alfredo Rossi, Wolfgang Funk, Tamás Bíró, Ralf Paus, Jens Seibel, Mira Jakobs, Tanju Yesilkaya, Janin Edelkamp, Marta Bertolini

**Affiliations:** 1Monasterium Laboratory Skin & Hair Research Solutions GmbH, 48149 Münster, Germany; 2Mediteknia Hair Transplant Clinic and Hair Lab, Universidad Fernando Pessoa Canarias, Gran Canaria, Canary Islands, 35450 Guía, Spain; 3Department of Clinical Internal Anesthesiological and Cardiovascular Sciences, “Sapienza” University of Rome, 00161 Rome, Italy; 4Schoenheitsklinik Dr. Funk, 81739 Munich, Germany; 5Dermatology & Cutaneous Surgery, University of Miami Miller School of Medicine, Miami, FL 33136, USA; 6Bayer Vital GmbH, 51373 Leverkusen, Germany

**Keywords:** female pattern hair loss, metabolism, nutrient supplementation, hair follicle

## Abstract

Female pattern hair loss (FPHL) is a non-scarring alopecia resulting from the progressive conversion of the terminal (t) scalp hair follicles (HFs) into intermediate/miniaturized (i/m) HFs. Although data supporting nutrient deficiency in FPHL HFs are lacking, therapeutic strategies are often associated with nutritional supplementation. Here, we show by metabolic analysis that selected nutrients important for hair growth such as essential amino acids and vitamins are indeed decreased in affected iHFs compared to tHFs in FPHL scalp skin, confirming nutrient insufficiency. iHFs also displayed a more quiescent metabolic phenotype, as indicated by altered metabolite abundance in freshly collected HFs and release/consumption during organ culture of products/substrates of TCA cycle, aerobic glycolysis, and glutaminolysis. Yet, as assessed by exogenous nutrient supplementation ex vivo, nutrient uptake mechanisms are not impaired in affected FPHL iHFs. Moreover, blood vessel density is not diminished in iHFs versus tHFs, despite differences in tHFs from different FPHL scalp locations or versus healthy scalp or changes in the expression of angiogenesis-associated growth factors. Thus, our data reveal that affected iHFs in FPHL display a relative nutrient insufficiency and dormant metabolism, but are still capable of absorbing nutrients, supporting the potential of nutritional supplementation as an adjunct therapy for FPHL.

## 1. Introduction

Female pattern hair loss (FPHL) is one of the most common causes of hair loss in women. The prevalence is known to increase with age and is estimated to be >25% in women above 49 years of age [1], which actually underestimate the prevalence of mild forms of FPHL [2,3,4,5]. While not life-threatening, FPHL can severely affect the psychological well-being of those experiencing it [6,7]. Between 70 and 88% of females with FPHL report that hair loss has negatively influenced their quality of life; accordingly, these women are more likely to have worse self-esteem and feelings of a negative body image [8,9]. 

By definition, FPHL is a non-scarring form of alopecia, characterized by a diffuse distribution of hair loss on the scalp [10], which exhibits several important differences to male pattern androgenetic alopecia and must be carefully distinguished from the latter [2,3,4,5]. FPHL results from an abnormally high percentage of non-fiber producing telogen hair follicles (HFs) and the progressive replacement of terminal (normal) (t)HFs with intermediate/miniaturized (i/m)HFs [2,11,12,13,14], which produce much thinner or invisible hair shafts [15]. As a consequence, reduction in hair density is initially visible in the parting hair line in the centro-parietal scalp, which can then extend to include the frontal line, and bi-temporal scalp regions with disease worsening [1,10,12,16]. Unlike in men, hair thinning is observed also in the occipital scalp, yet to a lesser extent compared to other scalp locations [17]. Without treatment, FPHL is generally progressive, albeit the rate at which it does so is widely variable. Episodes of hair shedding are unpredictable and may be spaced by 3 to 12 months, contributing to the distress elicited by the condition [18].

Of note, an early diagnosis is important [2], as current treatments are more effective at avoiding the progression of hair loss and follicle miniaturization rather than promoting (lost) hair regrowth [1]. Three important avenues of therapeutic intervention exist: topical, systemic, and surgical [1,19]. Along with the hair growth promoter minoxidil [20], the majority of prescribed treatments act by interfering with the activity of androgens [1,21], despite their very unclear role in FPHL pathogenesis [3,13,22,23,24]. Nutritional supplementation is also often recommended as an adjunct therapy [11,25,26]. This is based on the observation that deficiencies in nutrients such as biotin, vitamin D, vitamin B12, iron and/or zinc ions, alanine etc., have been correlated with some forms of hair loss [27,28,29,30,31,32], including FPHL [33,34]. 

However, it remains unknown whether and how nutritional deficiencies occur within human FPHL-associated HFs. Here we attempted to elucidate this aspect of FPHL pathogenesis by investigating whether affected iHFs [15] and tHFs from the parietal and occipital scalp of clinically diagnosed FPHL patients are characterized by a different nutrient/metabolite profile by employing untargeted metabolomics analysis [35]. Hair shaft production occurring during the growing (anagen) phase of the HF cycle is intimately linked with cell proliferation [36], and metabolic processes [37,38,39], namely glutaminolysis and aerobic glycolysis [40,41,42] and mitochondrial respiration [43]. Thus, we also aimed at gaining insights into possible alterations in metabolic activity between tHFs and iHFs by assessing the consumption of glucose and glutamine, as well as the associated production of lactate and glutamate. Furthermore, we asked whether different HF types in FPHL patients exhibit differences in perifollicular vascularization and angiogenesis-associated growth factors, i.e., vascular growth factor (VEGF) and trombospondin-1 (TSP-1) using quantitative (immuno-)histomorphometry in biopsies from parietal and/or occipital scalp given that nutrient supply and hair growth depend on adequate angiogenesis [44]. Lastly, we sought to confirm that intermediate and terminal HFs from FPHL patients are capable of absorbing exogenously administered nutrients by systemic administration (into the medium) of selected metabolites in FPHL HF organ culture ex vivo.

Taken together, our data show that affected iHFs in FPHL display a relative nutrient insufficiency and relatively dormant HF metabolism, but are still capable to absorb nutrients.

## 2. Materials and Methods

### 2.1. Human Samples

Occipital and parietal human scalp skin or follicular units were obtained from six healthy donors (aged 24–56 years) undergoing cosmetic surgery and thirteen FPHL patients (aged 35–70 years) (Table 1) [45,46] after informed consent and ethical approval were obtained (University of Muenster, no. 2015-602-f-S; University “La Sapienza” Rome, n. 2973, 28-11-13, University of Fernando Pessoa Canarias 03/2020 (2020-06-22)). All experiments on human tissue were performed according to Helsinki guidelines.

### 2.2. UPLC-MS

For metabolomics analysis, directly after surgery, human anagen VI scalp HFs from the occipital and the parietal scalp regions of *n* = 3 FPHL patients were microdissected and measured. For each donor, based on hair bulb diameter, hair shaft diameter, hair follicle length (Appendix A) and scalp region, the hair follicles were pooled in four experimental groups (terminal (t) and intermediate/miniaturized (i/m) HFs [15] from occipital and parietal scalp skin), and snap frozen in liquid nitrogen. Untargeted metabolic analysis was outsourced to Creative Proteomics Inc. (Shirley, New York, NY, USA). After thawing the samples on ice, 80% methanol was added and samples were homogenized twice on a MM 400 mill mixer at 60 Hz for 2 min, vortexed for 60 s, and sonicated for 30 min at 4 °C. Samples were then incubated at −40 °C for 1 h, vortexed for 30 s, incubated for 30 min at 4 °C, and centrifuged at 12,000 rpm for 15 min at 4 °C. Supernatants were kept for 1 h at −40 °C, centrifuged at 12,000 rpm and 4 °C for 15 min, and lyophilized to dryness. The pellets were resuspended in 100 μL of 80% methanol and 2.5 μL of internal standard (140 μg/mL, DL-o-chlorophenylalanine). After centrifugation at 12,000 rpm and 4 °C for 15 min, supernatants were subjected to LC-MS analysis (Thermo Scientific, Waltham, MA, USA, Ultimate 3000LC, Q Exactive). Untargeted metabolomic analysis was performed separately for each donor. For each identified peak, the relative abundance (peak area) is normalized to the sum of all peak areas within each sample (to the total ion current (TIC) for each sample) [35,47].

### 2.3. Metabolite Enrichment Analysis

Metabolite set enrichment analysis was performed using the web-based tool MetaboAnalyst 5.0 (University of Alberta, Edmonton AB T6G 2E8, Canada) (https://www.metaboanalyst.ca/MetaboAnalyst/home.xhtml, accessed on 30 March 2022) using KEGG as metabolite set library. Enrichment ratio is computed by observed and expected hits as described in [48,49].

### 2.4. Metabolic Activity Ex Vivo

For metabolites consumption/secretion analyses ex vivo, anagen VI terminal occipital HFs from *n* = 3 healthy female donors and occipital and parietal, terminal and i/m anagen VI HFs (Appendix A) [15] from *n* = 4 FPHL patients were microdissected and cultured for 24 h at 37 °C with 5% CO_2_ in a minimal media of William’s E media (Gibco, Life Technologies, Carlsbad, CA, USA) supplemented with 10 ng/mL hydrocortisone (Sigma Aldrich, St. Louis, MO, USA), 10 μg/mL insulin (Sigma Aldrich) and 1% penicillin/streptomycin mix (Gibco) to make Williams Complete Media (WCM) modified as previously described [46,50]. Day 1 culture media were tested with Glo™ Assays kits (Promega^®^, Madison, WI, USA) for glucose, lactate, glutamate, and glutamine/glutamate metabolites following the recommended protocol. All samples were diluted 100× in PBS.

### 2.5. Ex Vivo Absorption of Fluorescently Labeled Metabolites

Full length terminal and i/m occipital and parietal anagen VI HFs (Appendix A) [15] from *n* = 4 FPHL patients were microdissected and individually cultured for 24 h ex vivo at 37 °C with 5% CO_2_ in WCM medium. After a 3 h starvation period in PBS, the hair follicles were cultured for 1 h at 37 °C with 5% CO_2_ in WCM medium containing a combination of three fluorescent labeled metabolites as follows: 0.25 mM L-Cystine-Cy3 + 0.6 mM glutamic acid-Cy5 + 0.2 mM pantothenic acid-FAM. The culture was terminated by embedding the HFs in OCT and freezing them in liquid nitrogen. A total of 6 μm cryosections were obtained with a Leica cryostat. Images were taken using a Keyence fluorescence microscope BZ9100 (Keyence, Osaka, Japan) maintaining a constant set exposure time throughout imaging.

### 2.6. Immunofluorescence In Situ

For in situ investigations, punch biopsies of intact skin were directly frozen after explant in OCT embedding medium (Thermo Fisher Scientific) and cryosectioned in 7 µm thick sections with a Leica cryostat [45,46,51].

#### 2.6.1. CD31/COLIV

Tissue cryosections were fixed in acetone, pre-incubated with 10% goat serum in TBS, and incubated with the primary antibodies (mouse anti-human CD31 (PECAM), DAKO. 1:30 and rabbit anti-human Collagen IV, Novus biologicals, 1:100) at 4 °C overnight. Secondary antibodies incubation was performed at RT for 45 min. Counterstaining with 4′,6-diamidino-2-phenylindole (1 µg/mL) was performed to visualize nuclei.

#### 2.6.2. VEGF/CD31

Tissue cryosections were fixed in acetone, pre-incubated with 10% of goat serum in TBS, and incubated with the primary antibodies (mouse anti–human CD31 (PECAM), DAKO. 1:30 and rabbit anti-human VEGF (PA5-16754, Thermo Fischer 1:50) at 4 °C overnight. Secondary antibodies incubation was performed at RT for 45 min. Counterstaining with 4′,6-diamidino-2-phenylindole (1 µg/mL) was performed to visualize nuclei.

#### 2.6.3. TSP-1/CD31 

Tissue cryosections were fixed in acetone, pre-incubated with 10% of goat serum in TBS, and then incubated with the primary antibody against human CD31 (DAKO. 1:30) at 4 °C overnight followed by secondary antibody incubation at RT for 45 min. Sections were then incubated with a primary antibody against human TSP-1 (Thermo Fisher. 1:50) at 4 °C overnight followed by secondary antibody incubation at RT for 45 min. DAPI (1 µg/mL) counterstaining was performed to visualize nuclei.

### 2.7. Quantitative (Immuno-)Histomorphometry

Images were taken using a Keyence fluorescence microscope BZ9100 (Keyence, Osaka, Japan) maintaining a constant set exposure time throughout imaging for further analysis. The number of CD31+ blood vessels lumen was counted in the perifollicular tissue and in the dermal papilla (DP) [52]. VEGF staining immunoreactivity was measured in the connective tissue sheath (CTS) at the bulge, sub-bulge, and bulbar HF regions and in DP while TSP-1 staining immunoreactivity was measured in the hair follicle epithelium, and CTS of bulge, and bulb HF regions (Appendix A) [51]. Analyses were carried out with ImageJ (National Institutes of Health, Bethesda, MD, USA).

### 2.8. Statistics

Statistical analyses were performed with Graph Pad Prism 9 (Graph Pad, San Diego, CA, USA). The normality of the distribution(s) of the data was analyzed with the D’Agostino & Pearson omnibus test. Whenever datasets followed a normal distribution, the parametric one-way ANOVA and Tukey’s multiple comparison and multiple unpaired Student’s *t*-test with Benjamini and Hochberg 5%FDR correction, were used. If the data were not normally distributed, the non-parametric Kruskal–Wallis test, followed by Dunn’s multiple comparison test and multiple Mann–Whitney test with Benjamini and Hochberg 5%FDR correction, were applied. Data are expressed as mean ± SEM; *p* values < 0.05 were considered significant as indicated in the figure legends.

## 3. Results

### 3.1. Intermediate HFs from FPHL Patients Show Nutrient and Metabolite Deficiency

In light of clinical reports stating that supplementation with certain nutraceuticals may increase terminal hair count, anagen rate, and hair shaft thickness in FPHL patients [25,26,53], we hypothesized that affected iHFs from patients with FPHL might show some degree of nutrient insufficiency. To this end, we obtained intermediate and terminal HFs from patients with FPHL. Intermediate HFs were identified and sorted based on previously published properties [15], namely a shorter length and smaller hair shaft, bulb, and DP diameters (Appendix A). An untargeted metabolic analysis, employing ultra-performance liquid chromatography–mass spectrometry (UPLC–MS) [35,47] was performed separately in iHFs and tHFs from the occipital (OCC) or parietal (PAR) scalp from FPHL patients (*n* = 4 experimental groups: tOCC, tPAR, iOCC, iPAR per patient, *n* = 3 patients). 

Between 144 and 345 metabolites were identified in the HFs from each FPHL patient regardless of the HF type (Figure 1a), also revealing that HFs from each patient had a unique nutrient/metabolite profile, most likely because of differences in diet, hormonal, and genetic background. Amongst the detected nutrients and/or metabolites, 81 were shared among all three patients, whilst 15, 16, 29 were shared between two patients (Figure 1a). Most importantly, the relative abundance of nutrient/metabolites detected in at least two out of three patients was different in terminal and intermediate HFs from parietal or occipital scalp from FPHL patients (Figure 1b). Namely, the relative abundance of several of the detected nutrient/metabolites was lower in iHFs when compared to that in tHFs from the same scalp location, with very few exceptions (Figure 2a). 

The differentially abundant molecules, when analyzed using the functional analysis module of the metabolomics data analysis software MetaboAnalyst 5.0 (University of Alberta, Edmonton AB T6G 2E8, Canada) (https://www.metaboanalyst.ca/MetaboAnalyst/home.xhtml, accessed on 30 March 2022) were associated with several anabolic and biosynthetic pathways broadly associated with cellular proliferation, such as aminoacyl-tRNA biosynthesis, glutamine and glutamate metabolism, biosynthesis of several amino acids, glycolysis, and citrate cycle (TCA cycle) (Figure 2b). Particularly, a lower abundance of citric acid and malic acid, products of the TCA cycle, and of lactic acid, resulting from aerobic glycolysis, was found in occipital and/or parietal iHFs along with an increase of L-glutamine in selected patients, which is usually metabolized to glutamate during glutaminolysis (Figure 2c). 

In addition, essential amino acids or molecules often found in nutricosmetics, such as pantothenic acid (vitamin B5), L-tryptophan, L-carnitine, and L-valine were reduced in iHFs, vs tHFs in both scalp sites (Figure 2c). In contrast, L-cysteine and L-alanine were decreased only in parietal iHFs (Figure 2c). 

Therefore, our data indicate that affected iHFs from FPHL patients reveal changes consistent with nutrient and metabolite insufficiency.

### 3.2. Intermediate HFs from FPHL Are Characterized by Dormant Metabolism

Given our metabolomics findings revealing decreased levels of lactate, the main product of aerobic glycolysis, as well as of citric acid and malic acid, products of the TCA cycle, along with enrichment of L-glutamine, the substrate of glutaminolysis (Figure 1b and Figure 2b,c), we hypothesized that the overall cellular metabolic status of FPHL iHFs would be different than that of tHFs. To study the differential metabolic profiles, we organ-cultured occipital tHFs from “clinically” healthy donors and occipital and parietal terminal and intermediate HFs from FPHL patients and measured the concentration of glucose, lactate, glutamine, and glutamate in the medium after one day of ex vivo culture. 

First, we compared the concentrations of the different metabolites in the culture medium of occipital healthy and FPHL tHFs. While the release of lactate (Appendix A) and consumption of glucose (Appendix A) or glutamine (Appendix A) was relatively similar between occipital tHFs obtained from healthy donors and FPHL patients, glutamate secretion was significantly reduced in the latter (Appendix A), suggesting overall metabolic differences in healthy and FPHL HFs, in particular lowered glutaminolysis. 

Interestingly, lactate secretion into the medium was significantly decreased by both occipital and parietal FPHL iHFs when compared to their terminal counterparts (Figure 3a). This decreased lactate secretion could not be solely explained by decreased glucose uptake, as the extracellular glucose concentration was either similar or even higher in the supernatants of FPHL tHFs (Figure 3b). The rates of glutamine uptake were also similar between FPHL terminal and intermediate HFs, independently of the scalp region from which those HFs were obtained (Figure 3c). Despite having similar rates of glutamine consumption, FPHL tHFs secreted glutamate to the extracellular space, as shown by the increase of glutamate concentration in the supernatants of these cultures when compared to unconditioned medium (Figure 3d). In contrast, the glutamate concentration in the supernatant of iHFs was very close to the basal concentration of the culture medium (Figure 3d). 

Taken together, our results suggest that iHFs from patients with FPHL are characterized by a lower degree of aerobic glycolysis and glutaminolysis.

### 3.3. Intermediate Parietal HFs Display Lower VEGF and TSP-1 Expression, but TSP-1 Expression Is Significantly Increased in FPHL versus Healthy Terminal HFs

Because growing HFs produce and secrete vascular endothelial growth factor (VEGF) in the surrounding tissue to enhance vascularization providing nutrients [44,54,55,56], we assessed whether VEGF was differentially expressed in FPHL HFs vs healthy controls by measuring it by quantitative (immuno-)histomorphometry in the connective tissue sheath (CTS) surrounding the bulb, bulge, and sub-bulge regions, as well as in the DP (Appendix A). Of note, we could identify very few occipital iHFs in the patients analyzed, in line with their clinical phenotype of stage I and II Ludwig or Sinclair scales [57,58] (Table 1). Therefore, only terminal occipital and parietal terminal and intermediate HFs could be assessed.

Surprisingly, only a trend toward a decrease in mesenchymal VEGF expression was detected in occipital tHFs from FPHL patients in comparison to healthy donors (Appendix A). Occipital and parietal tHFs from FPHL patients revealed similar VEGF expression in the CTS, although parietal HFs had tendentially lower VEGF expression in the DP (Figure 4a,b). Most intriguingly, in the parietal FPHL scalp, diminished VEGF expression was found in iHFs as compared to tHFs, significantly in the sub-bulge CTS and tendentially in the bulge CTS (Figure 4b). 

VEGF expression and angiogenesis can be inhibited by thrombospondin 1 (TSP-1) [59,60,61]. Therefore, we hypothesized this mediator could be upstream of the lower VEGF expression in patients with FPHL. Surprisingly, lower expression of TSP-1 was seen in iHFs versus tHFs in the parietal scalp, even to a significant extend in the bulge CTS (Figure 4c,d). However, the expression of TSP-1 was increased in the bulb, bulge CTS, as well as bulge, and sub-bulge epithelia of occipital tHFs from patients with FPHL when compared to occipital healthy HFs (Appendix A). In addition, TSP-1 expression was even tendentially higher in the bulb CTS and sub-bulge epithelium of tHFs from parietal as compared to the occipital scalp (Figure 4c,d). 

Taken together, our data show that VEGF expression is decreased in parietal iHFs from FPHL patients. Despite a lower TSP-1 expression in parietal FPHL iHFs, our data also reveal the interesting observation that intrinsic differences may exist in the level of TSP-1 in occipital tHFs from FPHL patients and healthy donors, and also comparing tHFs from the relatively unaffected occipital scalp and hair thinning predisposed parietal scalp follicles in FPHL patients. Thus, our findings further point toward differences in expression of modulators of perifollicular vascularization associated with the progression of FPHL.

### 3.4. Perifollicular Vascularization Is Relatively Unchanged in Parietal Intermediate HFs, but Is Significantly Decreased in Parietal versus Occipital Terminal FPHL HFs

Given that parietal iHFs from patients with FPHL were characterized by different expressions of VEGF and TSP-1, we postulated that perifollicular vascularization may be impaired in intermediate HFs, thus explaining the decreased metabolic activity and intrafollicular nutrient/metabolite abundance. To probe this hypothesis, we quantified the area of CD31^+^ blood vessel cross sections (lumina) in the perifollicular CTS of intermediate and terminal HFs from different scalp regions of patients with FPHL and healthy donors (Appendix A), as an indicator of vascular density. Because there were so few iHFs in the occipital scalp skin biopsies of FPHL patients, we focused our attention on the parietal region. Interestingly, no difference in the CD31+ lumina/area in the DP was seen across the HF types, and scalp location in FPHL patients or healthy donors (Appendix A). Whilst we observed a similar density of perifollicular CD31+ lumina/area in occipital tHFs from FPHL and healthy donors (Appendix A), this was significantly decreased in tHFs from parietal as compared to the occipital scalp from FPHL patients (Figure 5a,b). Yet, only tendentially fewer CD31+ lumina per area were detected in the CTS in intermediate versus terminal HFs in the parietal scalp from FPHL patients (Figure 5a,b).

These data suggest that intrinsic differences are present between the vascularization of occipital versus parietal HFs, and that iHFs do not show significant impairment in perifollicular vascularization when compared to their terminal counterparts.

### 3.5. FPHL HFs Can Retrieve Exogenously Supplied Nutrients

Having established that iHFs from patients with FPHL did not exhibit vascularization abnormalities but lower relative abundance of specific nutrients, we considered that the nutrient uptake mechanisms [62] may be less efficient and questioned whether deficiencies in specific nutrients/metabolites could be replenished through exogenous administration of a nutraceutical. As a proof-of-principle for exploring translational hair health benefits, we supplemented terminal and intermediate HFs with nutrients which are found in some hair growth supplements, namely L-cystine, pantothenic-, and glutamic acids [53,63,64,65,66,67]. To track the incorporation into HFs, we starved the HFs ex vivo for 3 h in PBS, followed by a 1-h incubation with the fluorescently labeled nutrients. 

Through fluorochrome detection, we revealed that iHFs were able to take up those exogenously supplied nutrients, at a similar or faster rate than tHFs (Figure 6a–c). Therefore, it appears that the nutritional deficiencies observed in iHFs (Figure 1a–c and Figure 2a,b) are correctable in principle, at least under human scalp HF organ culture conditions.

## 4. Discussion

Conventional treatment for FPHL and other forms of alopecia are often associated with nutraceutical supplementation, but scientific causal and mechanistic indications supporting this strategy are still poorly understood. In the present study, we present evidence that in patients with FPHL, the metabolic profile and metabolism of iHFs are different from that of tHFs. Specifically, affected iHFs are characterized by nutrient and metabolite insufficiency which highlight a relatively quiescent metabolic activity. In an attempt to clarify the mechanisms involved, we reveal that changes in perifollicular vascularization and thus access to blood-carried nutrients does not appear to be a contributory factor. However, we detected differences in the production and release of two important growth factors associated with angiogenesis and hair cycle, VEGF and TSP-1, between intermediate and terminal HFs from FPHL patients. Intriguingly, we also identified reduced perifollicular vascularization in parietal versus occipital tHFs from FPHL patients, which may explain the higher susceptibility of parietal scalp. Finally, we demonstrate that nutritional deficiencies in iHFs might be compensated by administration of exogenously supplied nutrients, placing nutraceuticals as ideal adjuvant in lessening hair loss and the progression of HF miniaturization in patients.

During the growth phase, HF keratinocytes in the hair matrix undergo high rates of proliferation and cell division to produce the hair shaft [36]. The high energy levels required, are provided by the catabolism of glucose to lactate, i.e., aerobic glycolysis [40,41,42,68], which culminates with lactate release [69,70]. Indeed, we confirm that organ cultured tHFs from both healthy donors and FPHL patients engage in aerobic glycolysis, illustrated by a high rate of lactate production. However, we found that both occipital and parietal iHFs of FPHL patients showed lower abundance of lactic acid compared to location-matched tHFs and released less lactate in organ culture, suggesting lower rates of aerobic glycolysis in affected iHFs. Aerobic glycolysis generates pyruvate, a substrate for the TCA cycle [71]. Since aerobic glycolysis is decreased in iHFs from FPHL patients, they might produce less pyruvate, resulting in a reduced activity of the TCA cycle, which could, in turn, explain the low amounts of citric- and malic acid [72] detected in either parietal or occipital iHFs. These findings underline the more quiescent metabolic profile of iHFs in comparison to tHFs.

Contrary to the expectations, we found increased glucose concentrations in the medium of parietal tHFs as compared to occipital tHFs. This might be a consequence of glycogenolysis, another important metabolic process occurring in the ORS of HFs [40]. It is well-known that glycogen is stored and metabolized into glucose, to provide immediate energy within HFs [73,74]. Therefore, excessive glycogenolysis might be utilized by susceptible, tHFs of FPHL patients, to compensate for the beginning quiescent metabolic activity, and could serve as an early marker of pathophysiological changes in HFs preceding development of FPHL.

The amino acid L-glutamine is a critical fuel for HF regeneration [75] and its absence in the culture medium causes premature catagen induction during organ culture [42]. In line with this, consumption of glutamine from the culture medium was observed during organ culture of HFs from both healthy donors and FPHL patients, with no obvious differences detectable. Yet, iHFs appear to tendentially require more glutamine than their terminal counterparts, which was also underlined by an enrichment of glutamine in intermediate versus terminal HFs from FPHL patients as detected in the metabolomic analysis. Despite the higher need of glutamine by iHFs, glutaminolysis, the process by which glutamine is metabolized to glutamate [76], appears to be reduced. Indeed, the abundance of L-glutamic acid, the acidic form of glutamate, and its secretion into the medium are lower in intermediate compared to terminal HFs from selected FPHL patients. While the role of glutamate for human hair growth remains elusive, a recent murine study has shown that topical application of glutamic acid to the skin causes increased hair growth, vascularity, and HF cell proliferation [65].

Interestingly, glutamate secretion into the extracellular space occurs concomitantly with the cellular import of L-cystine via an antiporter [66,77]. Intracellularly, L-cystine is converted to cysteine, a major component of keratins and a precursor of the anti-oxidant glutathione [77,78]. Therefore, reduced glutamate secretion in iHFs may result into a decreased L-cystine uptake. In line, we observed lower abundance of L-cystine in parietal intermediate HFs. Intriguingly, exogenous cystine supplementation has been proposed to have beneficial effects for hair growth [64,66].

The overall slower metabolism of iHFs from FPHL patients is also reflected by the lower abundance of essential amino acids and vitamins, including pantothenic acid. Pantothenic acid is a precursor to coenzyme A and is known to promote keratinocytes, fibroblast, and DP cell proliferation [79,80,81]. In mice, insufficient dietary pantothenic acid can lead to alopecia [82,83], and two clinical trials have suggested that oral pantothenate increases hair thickness in women with self-perceived hair loss [84,85]. Therefore, reduced pantothenic acid levels might impair hair growth.

Taken together, our data suggest that the detected nutrient deficiency in iHFs from FPHL patients may result from a slower metabolism in comparison to tHFs. Given the importance of an active metabolism to maintain the high rate of proliferation required for hair growth, and to provide the metabolites necessary for hair shaft production, it is conceivable that these defects may be involved in HF miniaturization in FPHL patients.

The bulge region of the HF is permanently neighbored by a vascular structure termed upper venule annulus [50,51,52], and the perifollicular blood vessels around the proximal bulb provide nutrients to the actively dividing and differentiating HF matrix cells during anagen, thus also contributing to HF size [44,54]. It has been suggested that the regression of perifollicular vascularization reduces the supply of nutrients to the HF and impairs the proliferation of hair matrix cells [54,86]. Furthermore, a reduced blood supply to the HF has been linked to HF miniaturization in hair loss disorders such as androgenetic alopecia [87,88,89], and treatments promoting vasodilation and angiogenesis, such as minoxidil, micro needling, and platelet rich plasma (PRP), are reported to be beneficial in these patients [90,91,92]. This is consistent with the notion that anagen tHFs rely on angiogenesis to achieve optimal hair follicle size, hair growth rates, and depth in the adipose tissues [44,54,93]. However, a study reported no changes between CD31^+^ blood vessel density between parietal scalp skin from healthy controls and FPHL patients [94]. Therefore, we aimed to clarify whether skin vascularization is altered in affected HFs from FPHL patients. We focused on perifollicular blood vessels, to assess whether vascular changes were one of the reasons of the nutrient insufficiency. Although a direct comparison of occipital intermediate and terminal HFs was not possible (see explanation in M&M), we did not detect major differences in the perifollicular vascularization of parietal intermediate and terminal HFs. This is in line with the observation that the pro-angiogenic factor, VEGF, and the anti-angiogenic factor, TSP-1, were both found to be down-regulated in iHFs versus tHFs in parietal FPHL scalp. Thus, the quiescent metabolism and the nutrient deficiency that characterize iHFs are no consequence of poorer vascularization.

Yet, very intriguingly, our data suggest intrinsic differences regarding angiogenesis in HF of FPHL patients, which may explain the faster progression of hair loss and miniaturization, especially in the parietal region in FPHL. While decreased VEGF and increased TSP-1 expression were identified in FPHL tHFs versus healthy tHFs, perifollicular blood vessel density remained unaltered. Interestingly, VEGF and TSP-1 are not only involved in angiogenesis, but respectively also in maintaining HF size [54] and in modulating hair cycle in mice [86]. Thus, expression changes of VEGF and TSP-1 might be involved also in premature catagen development in tHFs from FPHL patients. In line, the most predispose parietal tHFs of FPHL patients were characterized by an increased TSP-1 expression, in all analyzed HF compartments apart from bulge epithelium, a decreased VEGF expression in the DP, and a significant reduction in perifollicular blood vessel density, when compared to occipital tHFs.

Finally, we show that the intrafollicular concentrations of selected nutrients and nutricosmetics could be increased by exogenous supplementation ex vivo, indicating that iHFs retain the intrinsic capacity to absorb nutrients from the surrounding environment, and that uptake mechanisms are not impaired in iHFs [62]. This suggests supplementation of nutraceuticals as an adjuvant strategy to improve metabolic activity of iHFs in FPHL.

## 5. Conclusions

Despite the preliminary nature of this study, we show for the first time that nutrient insufficiency is a characteristic of iHFs in patients with FPHL, which is not caused by impaired nutrient uptake or poor perifollicular vascularization, these findings highlight nutraceutical supplementation as an inexpensive and safe adjunct therapy for FPHL management.

## Figures and Tables

**Figure 1 nutrients-14-03357-f001:**
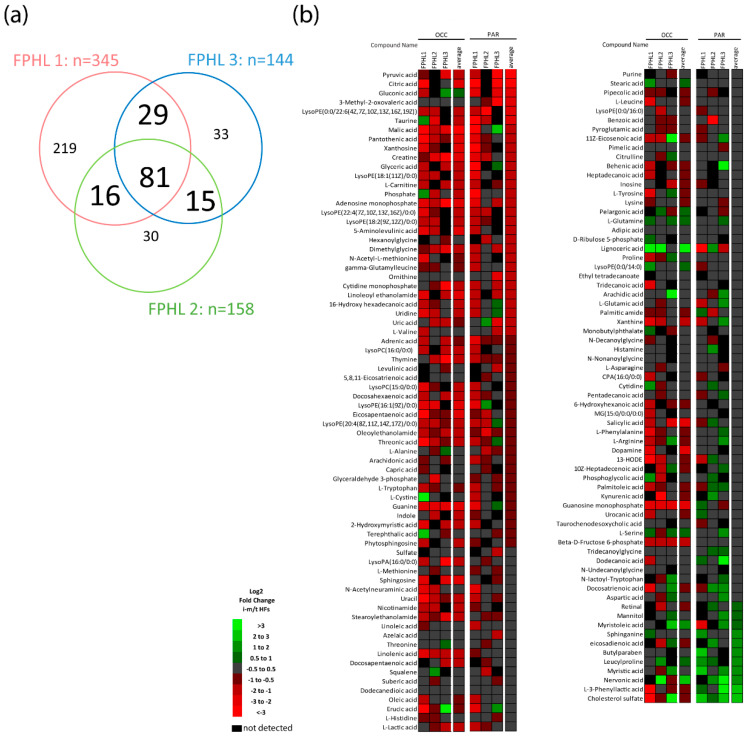
**Untargeted metabolomic analysis of terminal and intermediate hair follicles from FPHL patients.** (**a**) Number of metabolites identified by UPLC-MS analysis in *n* = 3−6 pooled terminal and *n* = 5–6 pooled intermediate/miniaturized hair follicles from the occipital and parietal scalp from *n* = 3 Female Pattern Hair Loss (FPHL) donors. *n* = 81 metabolites were identified in all three FPHL patients, *n* = 16 metabolites were commonly identified only in FPHL patients 1 and 2, *n* = 15 were commonly identified only in FPHL patients 2 and 3, and *n* = 29 were commonly identified only in FPHL patients 1 and 3. (**b**) Relative expression of the *n* = 141 metabolites identified in at least two FPHL patients. Heat maps show the Log2 fold change (FC) expression between intermediate/miniaturized and terminal hair follicles (i-m/t) in the occipital (OCC) and parietal (PAR) scalp region of each FPHL patient (FPHL1, FPHL2, and FPHL3) and the Log2 fold change (FC) average expression from the *n* = 3 FPHL donors.

**Figure 2 nutrients-14-03357-f002:**
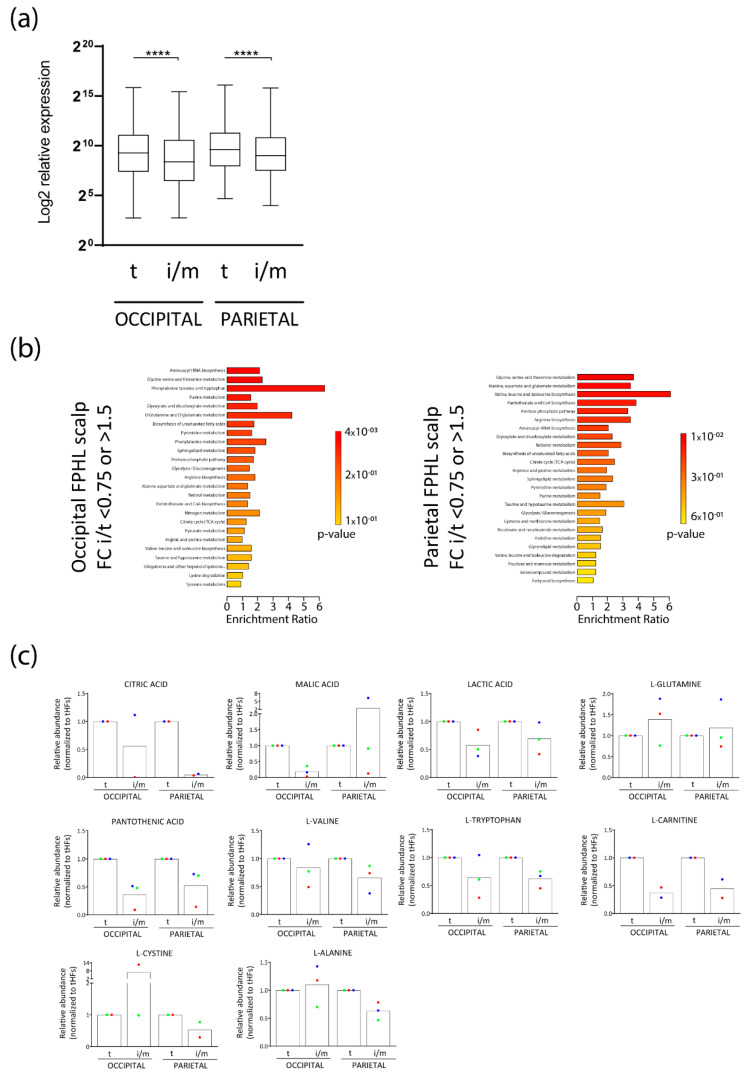
**Intermediate HFs from FPHL patients show metabolites deficiency.** (**a**) Log2 average relative expression of the *n* = 141 identified metabolites in at least two FPHL patients in terminal (t) and intermediate/miniaturized (i/m) hair follicles from occipital and parietal scalp skin samples from *n* = 3 FPHL donors. Box-plot with whiskers to minimum and maximum of the Log2 average relative expression data created with GraphPad9.0, *n* = 141, Friedman test *p* < 0.0001, Dunn’s multiple comparison test **** *p* < 0.0001. (**b**) Metabolite set enrichment analysis was performed using MetaboAnalyst 5.0 (University of Alberta, Edmonton AB T6G 2E8, Canada) (https://www.metaboanalyst.ca/MetaboAnalyst/home.xhtml, accessed on 30 March 2022) as basis differentially abundant (FC (i-m/t) < 0.75, FC (i-m/t) > 1.5) metabolites in intermediate/miniaturized versus terminal hair follicles in the FPHL occipital (**left**) and parietal (**right**) scalp region. (**c**) Relative abundance of selected metabolites in intermediate/miniaturized and terminal hair follicles from occipital and parietal FPHL scalp from *n* = 2–3 donors (green, red, and blue circles). Relative abundance normalized to corresponding terminal HFs.

**Figure 3 nutrients-14-03357-f003:**
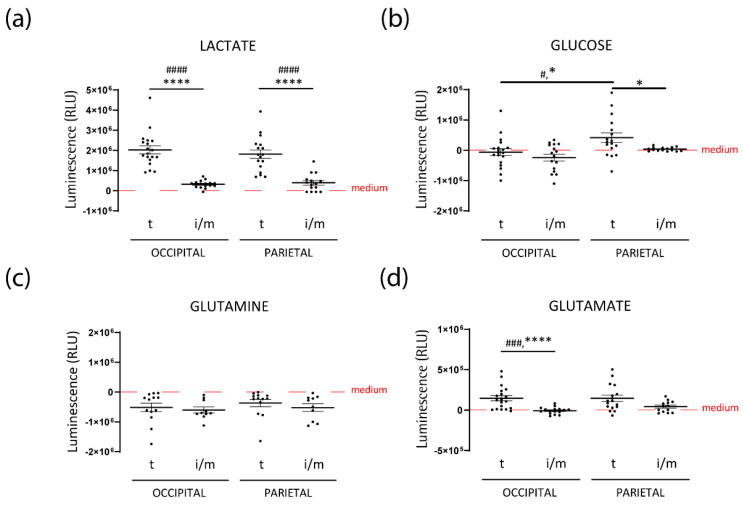
**Intermediate FPHL HFs reveal lower metabolic activity ex vivo.** (**a**) Lactate, (**b**) glucose, (**c**) glutamine, and (**d**) glutamate concentration was measured in the WCM culture medium after 24 h HF culture following recommended protocols for Glo™ Assays kits (Promega^®^, Walldorf, Germany). Mean ± SEM recorded luminescence from *n* = 9−19 HFs/ group from *n* = 3−4 FPHL donors. Red dotted lines indicate the amount in the blank, non-conditioned, WCM culture medium. GraphPad Prism 9.0; Kruskal–Wallis test and Dunn’s multiple comparison test # *p* < 0.05, ### *p* < 0.001, #### *p* < 0.00001; Multiple Mann–Whitney test (Benjamini and Hochberg 5%FDR correction), * *p* < 0.05, **** *p* < 0.00001. t: terminal; i/m: intermediate/miniaturized.

**Figure 4 nutrients-14-03357-f004:**
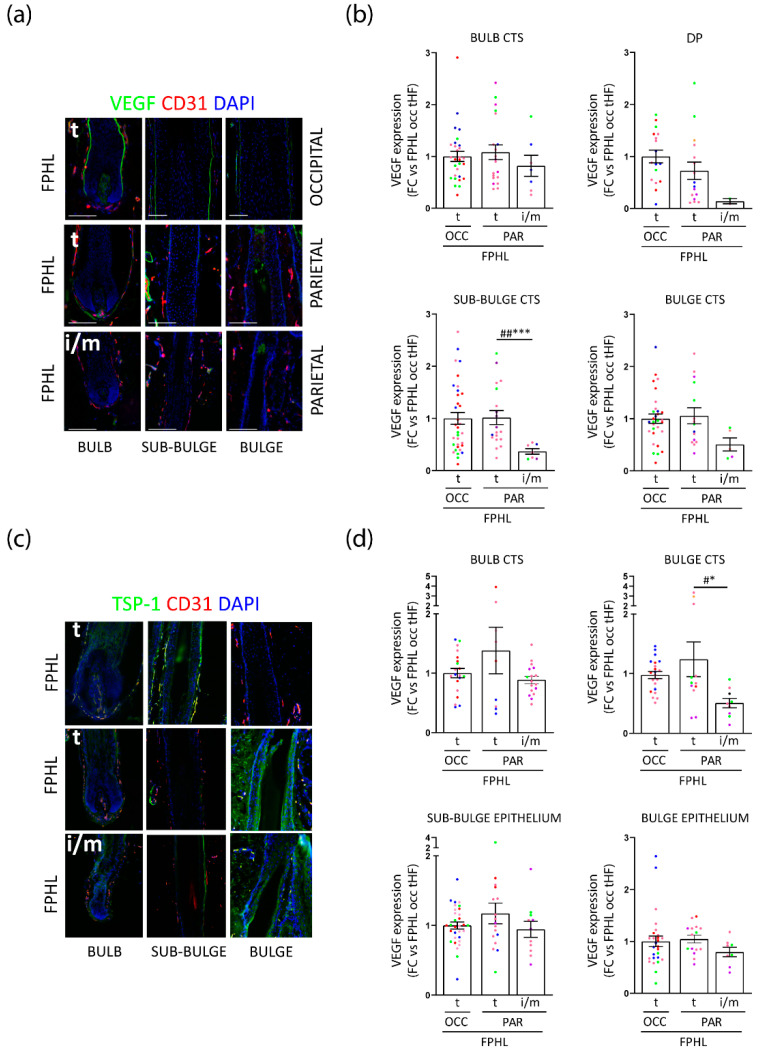
**Intermediate parietal HFs display lower VEGF and TSP-1 expression.** (**a**) Representative images of in situ Vascular Endothelial Growth Factor(VEGF)/Cluster of Differentiation 31 (CD31) (double immunofluorescence staining of the bulb, sub-bulge, and bulge areas of a terminal (t) hair follicle from FPHL occipital scalp skin and of a terminal, and an intermediate/miniaturized (i/m) hair follicle from FPHL parietal scalp skin. (**b**) VEGF protein expression levels were measured by quantitative (immuno-)histomorphometry in the connective tissue sheath (CTS) of hair follicle bulb, sub-bulge, and bulge regions, and in the dermal papilla (DP) of intermediate/miniaturized (i/m) and/or terminal (t) hair follicles from *n* = 4–5 FPHL donors (1: red, 2: green, 3: blue, 4: purple, 5: pink). Mean ± SEM from 2–34 hair follicles/group. GraphPad 9.0. Kruskal–Wallis with Dunn’s multiple comparison ## *p* < 0.01; Multiple Unpaired *t*-test or Mann Whitney test (Original FDR method -Benjamini and Hochberg −5%FDR correction), *** *p* < 0.001. (**c**) Representative images of in situ TSP-1/CD31 double immunofluorescence staining of bulb, sub-bulge, and bulge areas of a terminal hair follicle from FPHL occipital scalp skin and of a terminal, and an intermediate/miniaturized hair follicle from FPHL parietal scalp skin. (**d**) TSP-1 protein expression was measured by quantitative (immuno-)histomorphometry in the connective tissue sheath (CTS) of hair follicle bulb, and bulge and in the hair follicle epithelium at the sub-bulge and bulge level of intermediate/miniaturized (i/m) and/or terminal (t) hair follicles from *n* = 4–5 FPHL donors (1: red, 2: green, 3: blue, 4: purple, 5: pink). Mean ± SEM from 8–27 hair follicles/group. GraphPad 9.0. (CTS) Ordinary one-way Anova with Tukey’s multiple comparison test # *p* < 0.05; Multiple Unpaired *t*-test (Original FDR method -Benjamini and Hochberg −5%FDR correction), * *p* < 0.05; (epithelium) Kruskal-Wallis with Dunn’s multiple comparison, not significant (n.s.); multiple Mann Whitney test (Original FDR method -Benjamini and Hochberg −5%FDR correction), n.s.

**Figure 5 nutrients-14-03357-f005:**
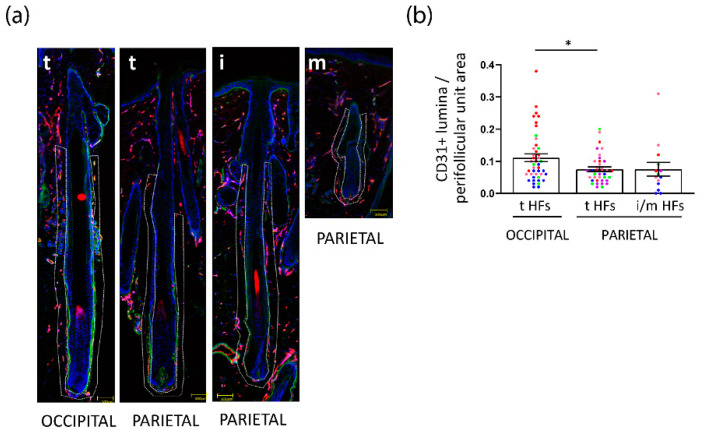
**Perifollicular vascularization is reduced in FPHL parietal scalp, yet not specifically in intermediate HFs.** (**a**) Representative images of in situ Cluster of Differentiation 31 (CD31)/Collagen IV (ColIV) double immunofluorescence staining of a terminal hair follicle from FPHL occipital scalp skin, and of a terminal, intermediate, and miniaturized hair follicle from FPHL parietal scalp skin. Dotted lines highlight the perifollicular areas. (**b**) The number of CD31+ lumina was counted in the perifollicular area of terminal and intermediate/miniaturized FPHL hair follicles mean ± SEM *n* = 14–43 HF from *n* = 4–5 FPHL donors (1: red, 2: green, 3: blue, 4: purple, 5: pink). Graph Pad Prism 9. Kruskal–Wallis test *p* = 0.0264, Dunn’s multiple comparison n.s.; multiple Mann–Whitney test with Benjamini and Hochberg −5%FDR correction, * *p* < 0.05.

**Figure 6 nutrients-14-03357-f006:**
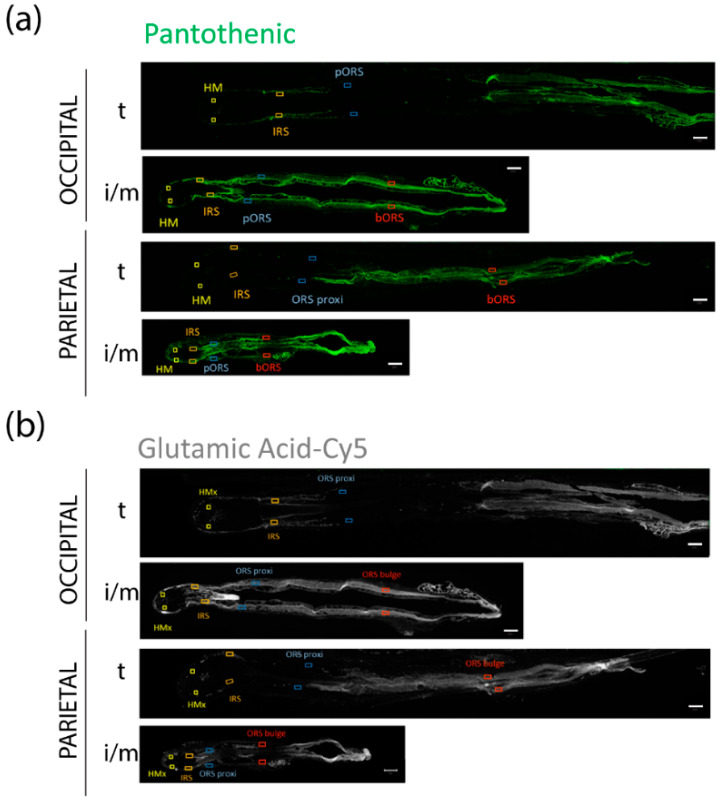
**Intermediate FPHL HFs are able to absorb fluorescent labeled metabolites ex vivo.** Representative images of terminal hair follicles (t) from FPHL occipital and parietal scalp skin and of intermediate/ miniaturized hair follicles (i/m) from FPHL parietal scalp skin. Hair follicles were imaged after three hours of starvation in PBS and one hour ex vivo culture in WCM medium supplemented with three fluorescent-labeled metabolites: pantothenic acid-FAM (**a**), glutamic acid-Cy5 (**b**), and L-cystine-Cy3 (**c**). Scale bars: 100 µM. HMx: hair matrix, IRS: inner root sheath; ORS: outer root sheath.

**Table 1 nutrients-14-03357-t001:** Overview of clinical samples used for the study.

Donors		Age	FPHL Grade *	Scalp Region	Source	Experiment
Healthy donors	1	45	-	occipital	skin	in situ immunostaining
	2	24	-	occipital	skin	in situ immunostaining
	3	40	-	occipital	skin	in situ immunostaining
	4	48	-	occipital	FUEs	metabolic activity
	5	54	-	occipital	FUEs	metabolic activity
	6	26	-	occipital	FUEs	metabolic activity
FPHL patients	1	50	I	occipital and parietal	skin	in situ immunostaining
	2	54	II	occipital and parietal	skin	in situ immunostaining
	3	62	II	occipital and parietal	skin	in situ immunostaining
	4	28	II	parietal	skin	in situ immunostaining
	5	60	I	occipital and parietal	skin	in situ immunostaining
	6	35	II–III	occipital and parietal	FUEs	UPLC-MS
	7	53	II	occipital and parietal	FUEs	UPLC-MS
	8	30	I	occipital and parietal	FUEs	UPLC-MS
	9	70	Ludwig III	occipital and parietal	FUEs	ex vivo absorption
	10	43	Ludwig II	occipital and parietal	FUEs	metabolic activity/ex vivo absorption
	11	65	Sinclair II	occipital and parietal	FUEs	metabolic activity/ex vivo absorption
	12	53	Sinclair II–III	occipital and parietal	FUEs	metabolic activity/ex vivo absorption
	13	42	Sinclair III	occipital and parietal	FUEs	metabolic activity/ex vivo absorption

* scales to clinically score FPHL disease severity; FPHL = Female pattern hair loss; FUEs = Follicular unit extractions; UPLC-MS = Ultra Performance Liquid Chromatography-Mass Spectrometry.

## Data Availability

Not applicable.

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
