# Peer review of "Intermediate Hair Follicles from Patients with Female Pattern Hair Loss Are Associated with Nutrient Insufficiency and a Quiescent Metabolic Phenotype"

_nutrients, 2022, doi:10.3390/nu14163357_

Round 1
Reviewer 1 Report
The researchers rightly point out that their work supports the hypothesis that affected iHF (intermediate scalp hair follicles) in FPHL (Female pattern hair loss) exhibit relative nutrient deficiency and dormant metabolism, but are still capable of absorbing nutrients. This suggests the potential of nutritional supplementation as an additional therapy for FPHL. From this point of view, it is an important work.
The analyses conducted suffer from low power due to the small study group. Researchers report the results of statistical tests that require correction of multiple comparisons without providing this correction, so as not to further reduce the power of the tests used.
I suggest performing the correction of multiple comparisons in accordance with the suggestions in paragraphs from (4) to (6) . If, however, the researchers decide to stay with the analyses carried out in this way and not to include the suggested corrections, then the limitations of this work, i.e. precisely the low power of the tests due to low counts, should be written in the discussion section and it should be explained that this was the reason for abandoning the correction of multiple comparisons.
(1) Line 223-225 - is functional analysis - a statistical method that should be given in the Material and Methods section in the description of the Statistics used.
(2) The Results section of the submitted paper presents far too many descriptions that should be in the introduction of the paper or in the discussion. The results section is a place to present your own results and not to introduce the topic or lead a discussion. In particular, please note the first paragraph of subsection 3.2, the first paragraph and partially the second paragraph of subsection 3.3, the first paragraph of subsection 3.4 and the first paragraph of subsection 3.5.
(3) Figure 2 (c): Please do not give the average and SEM for 3 measurements. The graphs indicate these 3 measurements and that is enough
(4) Figure 3: Comparisons were made by Kruskal-Wallis test and Dunn's test as a test of multiple comparisons. There is no need to provide the Mann-Whitney test, especially since it is given here without correction for multiple comparisons. If the Researchers nevertheless wish to provide the Mann-Whitney test in addition, please provide the p-value of this test with correction, e.g. I suggest the Benjamini-Hochberg FDR correction or with the Bonferroni correction. In the section describing the methods used, the description should then be supplemented accordingly.
(5) Figure 4 b) and d): Here again, if we use Student's t-test and Mann-Whitney test, they require correction of multiple comparisons, please add it or drop the results of these tests
(6) Figure 5: Comments as for Figure 4, but in addition:
Question: Why were outliers excluded "1 outlier removed ROUT (Q=0.1%)"? The statistical tests used i.e. Kruskal- Wallis and Mann-Whitney are based on ranks and are not sensitive to outlier measurements. Please provide a reason for excluding outlier measurements
(7) The abbreviation n.s. stands for "not significant" if we want to use this abbreviation it should at least be explained beforehand - this explanation is missing
Reviewer 2 Report
Dear authors.
After carefully reading your manuscript, I am with an excellent impression. A great development of the ideas, excellent the way you faced the hypothesis and developed it step by step during the results. The style used to expose the results and the working hypothesis of the sections and how you answered them, made the understanding of the results, and the integration of the ideas, to understand if the hypothesis initially raised was fulfilled.
I could only suggest to this manuscript, to associate references to the protocols used, as it would help to understand them in a better form, and would allow the reader to know the basis for the choice of these methodologies over others.
Excellent work.
Author Response
We thank Reviewer 2 for the positive revision. We have followed your suggestion, and added additional references in the Material and Methods sections (text below in red).
I could only suggest to this manuscript, to associate references to the protocols used, as it would help to understand them in a better form, and would allow the reader to know the basis for the choice of these methodologies over others.
2.1. Human samples: [Edelkamp et al., Methods Mol Biol. 2020; Bertolini et al., Plos One 2014]
2.2 UPLC-MS: [Miranda et al. Br J Dermatol 2010, AlGhamdi et al., Nutrients 2020, Roux et al. Clin Biochem 2011]
2.3 Metabolite enrichment analysis: [Pang et al., Nature Protocols 2022, Chong et al., Current Protocols in Bioinformatics 2019]
2.65. Immunofluorescence in situ: [Edelkamp et al., Methods Mol Biol. 2020; Bertolini et al., Plos One 2014; Piccini et al., Br J Dermatol. 2022 ]
2.7. Quantitative (immuno-)histomorphometry: [Keren et al., Sci Adv. 2022; Piccini et al., Br J Dermatol. 2022]